# NR2F1 Regulates TGF-β1-Mediated Epithelial-Mesenchymal Transition Affecting Platinum Sensitivity and Immune Response in Ovarian Cancer

**DOI:** 10.3390/cancers14194639

**Published:** 2022-09-24

**Authors:** Qiuju Liang, Zhijie Xu, Yuanhong Liu, Bi Peng, Yuan Cai, Wei Liu, Yuanliang Yan

**Affiliations:** 1Department of Pharmacy, Xiangya Hospital, Central South University, Changsha 410008, China; 2Department of Pathology, Xiangya Hospital, Central South University, Changsha 410008, China; 3National Clinical Research Center for Geriatric Disorders, Xiangya Hospital, Central South University, Changsha 410008, China

**Keywords:** ovarian cancer, platinum resistance, NR2F1, TGF-β1, EMT, CAF

## Abstract

**Simple Summary:**

In ovarian cancer (OC), platinum-based therapy remains the front-line therapy, but drug resistance is common. In the current study, we analyzed two Gene Expression Omnibus (GEO) datasets to identify the responsible genes involved in the mechanism of platinum resistance. Thirteen co-upregulated genes and one co-downregulated gene were obtained. Among them, NR2F1 revealed the highest correlation with a poor prognosis. Mechanism research revealed that NR2F1 promotes epithelial-mesenchymal transition (EMT) through TGFβ-1 signaling, participating in platinum resistance. We also found that NR2F1 was positively correlated with the infiltration of immunosuppressive cancer-associated fibroblasts (CAFs). Higher expression of NR2F1 was correlated with a poorer response to immune check blockades including anti-PD-L1. In the future, by analyzing the expression status of NR2F1 and the effect of platinum and immunotherapy in the clinical setting, it is expected that NR2F1 will be established as an effective drug selection marker, guiding treatment selection in OC patients.

**Abstract:**

The mechanism underlying platinum resistance in ovarian cancer (OC) remains unclear. We used bioinformatic analyses to screen differentially expressed genes responsible for platinum resistance and explore NR2F1′s correlation with prognostic implication and OC staging. Moreover, Gene-set enrichment analysis (GSEA) and Gene Ontology (GO) analyses were used for pathway analysis. Epithelial-mesenchymal transition (EMT) properties, invasion, and migration capacities were analyzed by biochemical methods. The association between NR2F1 and cancer-associated fibroblast (CAF) infiltration and immunotherapeutic responses were also researched. A total of 13 co-upregulated genes and one co-downregulated gene were obtained. Among them, NR2F1 revealed the highest correlation with a poor prognosis and positively correlated with OC staging. GSEA and GO analysis suggested the induction of EMT via TGFβ-1 might be a possible mechanism that NR2F1 participates in resistance. In vitro experiments showed that NR2F1 knockdown did not affect cell proliferation, but suppressed cell invasion and migration with or without cisplatin treatment through the EMT pathway. We also found that NR2F1 could regulate TGF-β1 signaling, and treating with TGF-β1 could reverse these effects. Additionally, NR2F1 was predominantly associated with immunosuppressive CAF infiltration, which might cause a poor response to immune check blockades. In conclusion, NR2F1 regulates TGF-β1-mediated EMT affecting platinum sensitivity and immune response in OC patients.

## 1. Introduction

Ovarian cancer (OC) currently ranks fifth in cancer-related mortalities among women and accounts for more deaths relative to any other malignancy influencing the female reproductive system [1]. The standard treatment for OC involves combination strategies of cytoreductive surgery and platinum-based chemotherapy [2]. Cisplatin, the most effective platinum-based anticancer agent, is widely used in OC chemotherapeutic treatments [3]. Cisplatin mainly acts through binding covalently to the N7 positions of purine bases, resulting in DNA structural damage in cancer cells and the subsequent block in cell division and activation of the apoptotic program [4]. Oxidative stress is also considered a fundamental mechanism underlying cisplatin cytotoxicity, and the massive release of reactive oxygen species (ROS) contributes to the apoptotic pathway activation [5]. Although the treatments produce a favorable initial response rate of 60 to 80%, the majority of patients eventually become platinum-resistant with subsequent relapses [6]. Poor appreciation of the mechanisms underlying OC platinum resistance poses a significant challenge for OC treatment. Thus, investigation of the molecular contributors to platinum resistance and the development of novel targeted therapeutic options for OC are desperately needed.

The nuclear receptor subfamily 2 group F member 1 (NR2F1), a well-established orphan nuclear receptor, was first cloned in 1986 [7,8]. In particular, evidence from accumulative studies has suggested a direct influence of NR2F1 on cancer progression via modulating cancer cells’ ability to migrate, proliferate or respond to external signals, including hormones [9]. For example, NR2F1 facilitates metastasis, invasiveness and dormancy of salivary adenoid cystic carcinoma cells through CXCL12/CXCR4 pathway activation [10]. Enhanced expression of NR2F1 favors migration and invasion capabilities of MCF-7 breast cancer cells, which is related to the repression of E-cadherin expression and the amelioration of the MAPK-signaling pathway [11]. Intriguingly, NR2F1 plays dual roles in breast cancer to inhibit tumor proliferation but promote invasion, both of which cause the accumulation of disseminated tumor cells (DTCs) in metastatic organs [12]. Estrogen receptor α (ERα) mediates the response to female steroid hormones in OC [13]. NR2F1 could selectively regulate the transcriptional activity of ERα signaling and change tumor cells’ response to anti-estradiol treatments [9]. However, few studies have probed NR2F1′s role in chemotherapy resistance.

Ovarian cancer, which is highly metastatic, displayed epithelial to mesenchymal transition (EMT) features and was characterized by the loss of epithelial marker E-cadherin and the acquirement of mesenchymal markers vimentin and N-cadherin [14]. TGF-β1 has been illustrated to be overexpressed in OC, and a growing number of studies have proved that TGF-β1 expression could contribute to cancer stem cell properties and EMT in ovarian clear cell carcinomas [15]. Moreover, TGF-β1 could induce the production of collagen [16], an important molecule favoring cancer invasion and migration [17]. Researchers discovered that TGF-β1 knockdown could ameliorate cisplatin sensitivity of OC cells by elevating the breast cancer susceptibility gene 1 (BRCA1) levels and Smad3 phosphorylation [18]. Based on the background mentioned above, TGF-β1 signaling might be a promising pathway participating in platinum resistance by regulating the EMT process in OC. In this study, we found NR2F1 upregulated in OC platinum-resistant tissues from Gene Expression Omnibus (GEO) database. Bioinformatic analysis showed NR2F1 could independently predict an inferior prognosis in patients receiving platin, upregulated in cisplatin-resistant OC cells, and positively correlated with pathological stage and half-maximal inhibitory concentration (IC50) value of cisplatin. Gene-set enrichment analysis (GSEA) and Gene Ontology (GO) analysis suggested that the TGF-β1-mediated EMT process might be the potential mechanism of NR2F1. Furthermore, in-vitro experiments revealed that NR2F1 could increase TGF-β1 expression and subsequently enhance OC cells EMT to promote platinum resistance. Further immune infiltration analysis evidenced that NR2F1 expression exhibited positive relation to cancer-associated fibroblast (CAF) infiltration. Meanwhile, NR2F1 high expression was associated with poorer immunotherapeutic response. Our study suggested the potential for NR2F1 as a therapeutic target for platinum-resistant and immunotherapeutic-resistant OC patients.

## 2. Materials and Methods

### 2.1. Bioinformatics Analyses

Potential datasets from the GEO database (http://www.pubmed.com/geo, accessed on 26 March 2022) [19] were selected based on three inclusion criteria: (1) keywords: ovarian cancer and platinum resistance, (2) study type: expression profiling by array, and (3) attribute name: tissue. Two datasets associated with the platinum resistance of OC were identified, including GSE51373 [20] and GSE131978 [21]. GSE51373 included 12 platinum-resistant and 16 platinum-sensitive OC tissues, and GSE131978 contained 12 platinum-resistant and 11 platinum-sensitive OC tissues. Employing the GEO2R methods, we acquired the differentially expressed genes (DEGs) between the platinum-resistant versus sensitive tissues. All DEGs met the screening threshold (*p*-value < 0.05 and fold-change > 1.5). Venn analysis was performed via Omicstudio (https://www.omicstudio.cn/tool, accessed on 26 March 2022) to identify the common DEGs (co-DEGs). We also utilized heatmaps to visualize the expression data of co-DEGs between resistant and sensitive samples among the two datasets, separately.

To examine the prognostic impacts of co-DEGs in OC patients, the overall survival (OS) significance heatmap data and NR2F1’s survival plots were attained from the GEPIA2 (http://gepia2.cancer-pku.cn/#analysis, accessed on 20 May 2022) portal [22]. We further exploited the Xiantao tool (https://www.xiantao.love/, accessed on 30 June 2022) and Kaplan-Meier plotter (https://kmplot.com/analysis/, accessed on 30 June 2022) [23] to confirm the prognostic implications of NR2F1 both in OC patients receiving platinum or not. The prognostic index contained OS, disease-specific survival (DSS), progression-free survival (PFS), and post-progression survival (PPS). To further determine the impact of the NR2F1 expression in patients affected by OC, the univariate and multivariate Cox regression analyses were accomplished utilizing the Xiantao tool to test whether NR2F1 could predict prognosis in OC patients independently.

The NR2F1 expression pattern in different tissues, including normal ovary tissues, was analyzed through Human Protein Atlas (HPA) [24], Genotype-Tissue Expression (GTEx) [25], and Functional Annotation of the Mammalian Genome (FANTOM5) [26] datasets. Then, to explore how NR2F1 expression correlated with OC progression, we adopted tumor-immune system interaction database (TISIDB, http://cis.hku.hk/TISIDB/index.php, accessed on 30 June 2022) [27] to compare NR2F1 levels in OC patients at different clinical stages. The change in the NR2F1 expression at different stages of malignancy was simultaneously validated at the cellular levels based on GSE24789 [28], which included 3 mice ovarian surface epithelial (MOSE) early cell samples at a pre-neoplastic, non-malignant stage; 3 MOSE intermediate cell samples at a neoplastic, pre-invasive state; and 3 MOSE late cell samples at a malignant, invasive stage. Additionally, NR2F1 expression and specific genetic mutation related to OC were acquired from The Cancer Genome Atlas (TCGA) [29] and International Cancer Genome Consortium (ICGC) cohorts [30]. The expression difference of NR2F1 in BReast-CAncer susceptibility gene 1 (BRCA1), BRCA2, tumor protein p53 (TP53) and AT-rich interaction domain 1A (ARID1A) mutant patients and wild-type were visualized by boxplots. Further, we verified the NR2F1 mRNA levels in 3 pairs of cisplatin-resistant and cisplatin-sensitive OC IGROV-1 cell lines from the GSE58470 [31]. In the meantime, we explored the links between NR2F1 expression and cisplatin sensitivity of OC patients using Biomarker Exploration of Solid Tumors (BEST, https://rookieutopia.com/app_direct/BEST/#PageHomeAnalysisModuleSelection, accessed on 5 September 2022) portal. Finally, a chemotherapy-related dataset, GSE47856, was then used to investigate the impact of NR2F1 on the chemotherapy response of OC [32]. 

To determine the NR2F1-related phenotype and signal pathways, GSEA and GO analysis through the Xiantao tool was performed based on the DEGs between platinum-resistant and -sensitive groups in GSE51373 and GSE131978, respectively. Phenotypes showing adjusted *p* < 0.05 and false discovery rate (FDR) < 0.25 were considered significantly associated. Only phenotypes and signal pathways of interest were exhibited.

Tumor-infiltrating immunocytes linked with NR2F1 expression were investigated by exploiting Tumor Immune Estimation Resource 2.0 (TIMER2, http://timer.cistrome.org/, accessed on 2 July 2022) [33]. We firstly used XCELL algorithm to estimate how NR2F1 expression related to the abundances of CD8+ T-cell, B-cell, neutrophil, monocyte, macrophage, myeloid dendritic cell (DC), and natural killer (NK) cell, as well as CAFs with results shown as chord diagram. CAFs were chosen for an in-depth analysis applying the EPIC, MCPCOUNTER, and TIDE algorithms. Moreover, we also used HPA repository to visualize the expression of NR2F1 in distinct cell type clusters of ovary tissues, and the results were visualized by a UMAP plot and a bar plot. Additionally, based on the ESTIMATE algorithm [34], stromal scores were generated by taking advantage of the TCGA OC cohorts. Additionally, we assessed the link between NR2F1 and CAF markers in OC through GEPIA2 and TIMER2. In the end, RNA-sequencing profiles and detailed clinicopathological information for OC patients were extracted from TCGA and ICGC cohorts, and an immune-checkpoint blockade (ICB) response was predicted employing Tumor Immune Dysfunction and Exclusion (TIDE) algorithm [35]. Higher TIDE scores implied more resistance to ICB. Two immunotherapy datasets including IMvigor210 [36] and Wolf [37] were obtained from BEST to verify the effect of NR2F1 on patients’ response to anti-PD-L1. In parallel, the capacity of NR2F1 in distinguishing responders and nonresponders was evaluated through the receivers operating characteristic (ROC) curves analyses.

### 2.2. Cell Lines and Reagents

OC cell lines (A2780 and SKOV3), provided by the Center for Molecular Medicine at Xiangya Hospital, were placed in Dulbecco’s modified eagle medium (DMEM, Gibco, Billings, MT, USA) with 10% fetal bovine serum (FBS, Gibco, USA) and 1% penicillin-streptomycin (Gibco, USA) at 37 °C within a 5% CO_2_ sterile incubator. Cisplatin was purchased from Sigma. Recombinant human TGF-β1, derived from HEK293 cells, was purchased from R&D system (Cat No: 100-21, Minneapolis, MN, USA). 

### 2.3. Transfection of Cells

Negative control (NC) and si-NR2F1-1 (targeting sequence: AGCTTCAACTGGCCTTACA), si-NR2F2-2 (targeting sequence: GCAAGCACTACGGCCAATT) were purchased from Suzhou Ribo Life Science (Kunshan, China). The vectors and siRNAs were transfected, separately, into A2780 and SKOV3 cells utilizing lipofectamine 3000. Cells were stratified into (1) NC group, (2) si-NR2F1 group, and (3) si-NR2F1 group. To explore the role of NR2F1 in cisplatin sensitivity, we treated A2780 and SKOV3 cells with or without cisplatin 24 h after the transfection of siRNAs. Cisplatin was dissolved in DMSO (Sigma, Ronkonkoma, NY, USA). 

### 2.4. Cell Proliferation Assay

After 24 h of transfection, A2780 and SKOV3 cells were seeded in 96 well culture plates (2 × 10^3^ cells/well). Then, 20 μM cisplatin and a control medium were added for an additional 24 h. For dose-dependent experiments, the cells were separately cultured with cisplatin at 0, 10, and 20 μM for 24 h. CCK-8 test solution (Bimake, Houston, TX, USA) was used to detect cell proliferation in response to cisplatin. After one hour of incubating, cell viability was determined by recording the optical density values at a test wavelength of 450 nm utilizing a VICTOR X2 microplate reader (PerkinElmer, Waltham, MA, USA).

### 2.5. Western Blot (WB) Detection

Protein extracts of OC cells were prepared by applying a RIPA lysis buffer. The proteins were quantified employing a BCA kit, separated electrophoretically on 10% SDS-PAGE gels, electro-transferred to PVDF membranes, and blocked utilizing 5% skimmed milk. Afterwards, membranes were incubated with anti-NR2F1 (Cat NO: 24573-1-AP, Proteintech, Rosemont, IL, USA), TGF-β1 (Cat NO: 21898-1-AP, Proteintech, USA), E-cadherin (Cat NO: 20874-1-AP, Proteintech, USA), N-cadherin (Cat NO: 22018-1-AP, Proteintech, USA), and vimentin (Cat NO: 10366-1-AP, Proteintech, USA) primary antibodies with a dilution of 1:1000 (*v*/*v*). Next, the membranes were rinsed thrice in PBST for 3 min, incubated for 1 h with HRP-conjugated secondary antibodies (Cat NO: SA00001-1, Proteintech, USA), and finally, protein band intensities were assessed through the Image Lab software (Bio-Rad, Hercules, CA, USA). β-actin (1:1000; Santa Cruz, CA, USA) served as a normalized control. 

### 2.6. Assessment of Invasion via Transwell Assay

Matrigel was diluted in a serum-free medium to 0.1 mg/mL and immediately put into the apical chamber. The Matrigel was cured by incubating at 37 °C for 30 min. Prior to starting the assay, Matrigel invasion chambers were hydrated for 30 min. After a 24-h serum starvation, A2780 and SKOV3 cells were trypsinized and re-suspended in a serum-free medium at 5 × 10^5^ cells/mL. A total of 100 μL cell suspensions were loaded to the upper compartment whereas cold DMEM (600 μL) with 10% FBS was pipetted into the bottom chamber. Plates were subsequently incubated within a 5% CO_2_-contained atmosphere at 37 °C. On the next day, the transwell chambers were fixated utilizing 4% paraformaldehyde for 30 min before undergoing 0.1% crystal violet staining. Invasive cells were determined through cell counting within 5 randomly selected fields per well exploiting the inverted microscope. Taking the group with the largest amount of invading cells as 100%, the relative invasion rate was calculated as follows: relative invasion rate (%) = the number of invading cells in the treated group/the number of invading cells in the control group ×100%.

### 2.7. Wound Healing Assay

Cells were inoculated into 6-well-culture plates at a density of 1.2 × 10^6^ cells/mL. A 20 μL micropipette tip was adopted to generate a linear wound between cells via scratching the cell monolayers. The cells were then washed and grown in DMEM without FBS. Images were captured at 0 h and after 24 h with a microscope and the data were analyzed utilizing ImageJ Software.

### 2.8. qRT-PCR

Forty-eight hours after transfection, the RNA extraction was performed utilizing TRIzol reagent (Invitrogen, Waltham, MA, USA). Reverse transcription was conducted utilizing the PrimeScriptTM RT reagent kit (Takara, Beijing, China). Afterward, the PCR reaction was done with the following thermal program: 95 °C (30 s), 40 cycles of 95 °C (5 s), and 60 °C (30 s). Data were normalized to β-actin. The primers for TGFβ-1 are 5′-CTAATGGTGGAAACCCACAACG-3′ and 5′-TATCGCCAGGAATTGTTGCTG-3′. The primers for β-actin are 5′-CATGTACGTTGCTATCCAGGC-3′ and 5′-CTCCTTAATGTC ACGCACGAT-3′. The 2^−ΔΔct^ method was exploited to decide gene expression.

### 2.9. Immunohistochemistry

The OC tissue array (Cat NO: HOvaC154Su01) was acquired from Shanghai Outdo Biotech. The related clinicopathological details were also offered by this company. Immunohistochemical (IHC) was conducted as previously depicted [38]. Briefly, IHC was accomplished using Histomouse SP Kit (Invitrogen, USA). Sections were immunostained utilizing a streptavidin-peroxidase method following microwave antigen retrieval. 3,3′-diaminobenzidine was adopted to visualize positive signals. The antibodies against NR2F1 (Cat NO: 24573-1-AP) and platelet-derived growth factor receptor alpha (PDGFRA) (Cat NO: bs-0231R) were purchased from Proteintech. Two pathologists were invited independently to review and quantify the image of sections. IHC intensity score was subjectively ranked into four levels: 0 (negative), 1 (weak), 2 (moderate), or 3 (strong). Scores for staining extent were assigned as: 0 (≤10%), 1 (11–25%), 2 (26–50%), 3 (51–75%), or 4 (>75%). We calculated the final score through themultiplication of the two above-mentioned scores. Eventually, total scores > 1 were grouped as high expression, while those < 1 were classified as low expression.

### 2.10. Statistical Analyses

For two-group analyses, Student’s *t*-test was employed to determine statistical differences, and for analyses involving three or more groups, one-way ANOVA was employed. The log-rank test was adopted for comparing survival differences. All experiments were repeated thrice, with data reported as mean ± standard deviations (SD). Quantitative data were compared and graphically represented utilizing SPSS 26.0 and GraphPad Prism 8. Significant differences were accepted when *p* < 0.05.

## 3. Results

### 3.1. DEGs between Platinum-Resistant and Sensitive OC Patients

DEGs associated with platinum resistance were analyzed using gene expression profiles of two GEO datasets (*p* < 0.05 and FC > 1.5 as cutoff threshold). Using the GEO2R tool, platinum-resistant groups have total of 993 upregulated genes and 832 down-regulated genes in GSE51373, and 126 upregulated genes and 117 down-regulated genes in GSE131978 relative to platinum-sensitive groups (Appendix A). After analysis with a venn diagram, 13 co-upregulated platinum-resistant-related genes (FABP4, RHOBTB3, TIMP3, IGF1, IGFBP6, NR2F1, PALLD, CDH11, HSPA2, LTBP4, ACTA2, NAP1L1, and SNCA), and one co-downregulated gene (HIST1H2BD) were identified (Figure 1a,b). The heatmaps of co-DEGs in GSE51373 and GSE131978 are shown in Figure 1c,d. 

### 3.2. NR2F1 Predicts Dismal Prognosis in Ovarian Cancer

The prognostic significance of co-DEGs expression in OC patients was explored by applying the GEPIA2. As illustrated in Figure 2a, the expression of NR2F1 had the most significant relationship with OC patients’ OS (HR = 1.4, *p* = 0.0068), whereas others DEGs showed no obvious significance (*p* > 0.05) (Figure 2a,b). As well, patients with higher NR2F1 levels displayed inferior OS (HR = 1.48, *p* = 0.003) and DSS (HR = 1.36, *p* = 0.029) based on Xiantao tool (Figure 2c,d). Consistently, the Kaplan-Meier plotter demonstrated that the NR2F1 expression was prominently connected to shorter OS (HR = 1.27, *p* = 0.00022), PFS (HR = 1.2, *p* = 0.0045), and PPS (HR = 1.29, *p* = 0.003) in patients with OC (Figure 2e–g). Additionally, highly expressed NR2F1 also displayed poorer OS (HR = 1.32, *p* = 9.3 × 10^−5^), PFS (HR = 1.23, *p* = 0.0017), and PPS (HR = 1.29, *p* = 0.0033) in OC patients receiving platinum (Figure 2h–j). The univariate and multivariate Cox regression analyses were conducted for examining NR2F1’s ability to independently predict adverse OC prognoses. The results implied that NR2F1 was independently correlated with unfavorable OS after adjustment for stage, primary therapy outcome, race, age, grade, anatomic neoplasm subdivision, lymphatic and venous invasion, and tumor residual disease (Table 1). Taken together, these findings reveal that NR2F1 might act as an independent prognostic biomarker in OC.

### 3.3. NR2F1 Positively Correlates with Ovarian Cancer Development and Impacts the Treatment Outcomes of Ovarian Cancer

Through analysis of HPA, GTEx, and FANTOM5 datasets, a relatively high expression of NR2F1 was detected in normal ovary tissues compared to indicated organs (Appendix A). Furthermore, TISIDB and GSE24789 were further used to investigate the tendency of NR2F1 expression in OC at different clinical stages. The expression of NR2F1 in OC was marginally enhanced in advanced stages (Spearman’s r = 0.11, *p* = 0.0566) (Figure 3a). Similarly, NR2F1 levels of MOSE cells at a late stage were significantly increased than those of MOSE cells at an early stage (*p* < 0.05) (Figure 3b). The connection between NR2F1 levels and OC patients’ clinical parameters was also probed using the detailed clinical description of an OC tissue array. As shown in Table 2, the expression of NR2F1 exhibited a positive relation to pathologic stage (*p* < 0.001) as well as the T stage (*p* = 0.003), yet showed a negative relation to N (*p* < 0.001) and M stages (*p* = 0.001). Moreover, there was no significant relationship between NR2F1 expression and age (*p* = 0.054), ki67 intensity (*p* = 0.140) and extent (*p* = 0.057), and EGFR intensity (*p* = 0.614) and extent (*p* = 0.450). Additionally, NR2F1 expression and specific genetic mutation related to OC were acquired from TCGA and ICGC cohorts. Based on the TCGA database, no significant difference was observed in BRCA1 (*p* = 0.3512), BRCA2 (*p* = 0.6746), and TP53 (*p* = 0.9223) mutant patients compared to wild-type (Appendix A–c), while NR2F1 expression was significantly reduced in the ARID1A mutant group (*p* = 0.0238) biased with relatively small sample sizes (Appendix A). As for the ICGC dataset, no significant differences were observed in BRCA1 (*p* = 0.9626), BRCA2 (*p* = 0.5586), and ARID1A mutant (*p* = 0.9225) patients compared to wild-type (Appendix A–g). Then, decreased expression of NR2F1 was validated in cisplatin-resistant OC IGROV-1 cell lines using GSE58470 (*p* < 0.05) (Figure 3c). In parallel, the association of NR2F1 expression and cisplatin sensitivity was analyzed by the BEST portal. NR2F1 expression was positively correlated with IC50 values of cisplatin in OC related datasets GSE49997 and GSE13876 (both *p* < 0.05) (Figure 3d,e). Moreover, to further identify the effect of NR2F1 on the treatment outcomes of platinum, the expression levels of NR2F1 in GSE47856 associated with cisplatin chemotherapy were checked. Treatment with cisplatin downregulated NR2F1 expression in OC DOV13B, OVCA433 and C13 cell lines (all *p* < 0.05) (Figure 3f–h). The above findings indicated that NR2F1, a platinum resistance-related gene, was positively correlated with OC progression and might influence the chemotherapy response of OC.

### 3.4. GSEA and GO Analysis of DEGs in Platinum-Resistant Patients

The GSEA analysis of platinum-resistant-associated DEGs showed a significant correlation to the pathway of “multicancer invasiveness signature” in GSE51373 (*p* = 0.041) and GSE131978 (*p* = 0.030). Normalized enrichment scores for invasiveness were 1.927 and 2.706, respectively, indicating that EMT positively correlated with platinum resistance in OC (Figure 4a,b). Moreover, GO analysis demonstrated that DEGs were associated with a series of EMT-related pathways, covering transforming growth factor-beta receptor signaling pathways in GSE51373 and GSE131978 (both *p* < 0.05) (Figure 4c,d). The heatmap visualized the expression data of the DEGs in GSE51373 and GSE131978 enriched in the TGFβ receptor pathway. Additionally, as an essential player in inducing EMT, TGFβ-1 expression in platinum-resistant samples was found to be elevated in GSE51373 (Appendix A). In conclusion, our findings suggested that NR2F1 regulated platinum chemosensitivity through regulating TGFβ-1 participating in the EMT process.

### 3.5. Silencing NR2F1 Inhibits EMT

To assess the effect of cisplatin on NR2F1 expression, A2780 and SKOV3 cells were treated with 0–20 μM cisplatin. As shown in Appendix A, the NR2F1 protein levels decreased upon cisplatin treatment in a dose-dependent manner. Furthermore, the CCK-8 assay determined the dose dependence effect of 0–20 μM cisplatin on cell survival, while 20 μM cisplatin showed approximately half inhibitory effects and was selected for the subsequent experiments (Appendix A). To investigate whether NR2F1 affects the proliferation of cells’ response to cisplatin, CCK-8 assay was conducted to measure the cell viability without or with cisplatin treatment. As shown in Appendix A, silencing NR2F1 did not reveal a significant effect on cell viability in A2780 and SKOV3 cells.

As a critical step for supporting tumor invasion and migration, EMT has been validated as an essential pathway in OC platinum resistance [39]. To explore the function of NR2F1 in EMT signaling, multiple EMT markers, encompassing E-cadherin, N-cadherin, and vimentin after NR2F1 knockdowns were examined. As demonstrated in Figure 5a, siNR2F1-transfected A2780 and SKOV3 cells showed enhanced E-cadherin, as well as reduced N-cadherin and vimentin. To further evaluate NR2F1-mediated EMT upon platin treatment, wound-healing and transwell assays were exploited for checking the invasive and migratory capacities with or without cisplatin treatment. Knocking down NR2F1 repressed the invasion (Figure 5b–d) and migration (Figure 5e–h) in A2780 and SKOV3 cells, while the inhibitory effects were more significant in cisplatin treatment groups. These findings implied that NR2F1 might be responsible for resistance to platinum via the induction of EMT.

### 3.6. NR2F1 Mediated EMT via TGFβ-1

Considering that the TGF-β pathway serves a pivotal role in EMT progression and TGFβ-1 is a key modulator of the canonical TGF-β signaling cascades [40], we hypothesized that TGF-β1 may play an essential role in NR2F1-mediated resistance to platinum. To determine whether NR2F1 expression was linked with TGF-β1, the expression of TGF-β1 in NR2F1-silenced A2780 and SKOV3 OC cells was measured by adopting qRT-PCR and WB assay. As expected, decreased TGF-β1 mRNA expression was detectable in siNR2F1 transfected A2780 and SKOV3 cells relative to negative control (Figure 6a). As well, NR2F1 knockdown diminished the protein expression of TGF-β1 (Figure 6b). The effect of NR2F1 knockdown marked by E-cadherin, N-cadherin, and vimentin dysregulation was reversed by TGF-β1 treatment in A2780 and SKOV3 cells (Figure 6c,d). Moreover, treatment of exogenous TGF-β1 weakened the inhibitory effects of siNR2F1 on invasion (Figure 6e–g) and migration (Figure 6h–k). 

### 3.7. NR2F1 Correlates with CAF Infiltration and Immunotherapy Therapeutic Response

Evidence increasingly illustrates the relationship between TGF-β signaling and the immune microenvironment [41]. Therefore, the role of NR2F1 in the ovarian microenvironment was investigated. The coefficients of NR2F1 expression and immune cell infiltrations were calculated using the TIMER2.0 repository. According to the XCELL algorithm, immune-suppressive cells CAFs showed the highest relationship with NR2F1 (Figure 7a). Consistently, the expression of NR2F1 displayed a positive association with CAF infiltration after being adjusted by tumor purity based on the EPIC (Spearman’s r = 0.38, *p* = 5.96 × 10^−10^), MCPCOUNTER (Spearman’s r = 0.408, *p* = 1.97 × 10^−11^), and TIDE (Spearman’s r = 0.526, *p* = 4.37 × 10^−19^) algorithms (Figure 7b). Moreover, NR2F1 expression in ovary tissues was relatively high in fibroblast cell clusters (Figure 7c,d). NR2F1 expression exhibited a positive relationship with stromal scores (Spearman’s r = 0.2541, *p* < 0.0001) (Figure 7e). Correlation analyses across NR2F1 and markers for CAFs showed that NR2F1 expression was positively connected to PDGFRA (Spearman’s r = 0.56, *p* =1.4 × 10^−36^), PDGFRB (Spearman’s r = 0.46, *p* =4.1 × 10^−24^), alpha-smooth muscle actin (ACTA2) (Spearman’s r = 0.36, *p* =8.1 × 10^−15^), caveolin-1 (CAV1) (Spearman’s r = 0.36, *p* = 8 × 10^−15^), and fibroblast activation protein (FAP) (Spearman’s r = 0.22, *p* = 5.2 × 10^−06^) (Figure 7f–j). Results from TIMER2 further confirmed the positive relationship between NR2F1 and CAF markers (Appendix A). The CAF marker PDGFRA displayed the highest correlation in both portals. The representative IHC images demonstrated the strong NR2F1 and PDGFRA expression in metastases and weak expression in primary OC tissues (Figure 7k). The error bar plot visualized the overexpression of NR2F1 in metastatic OC tissues (*p* < 0.01) (Figure 7l). Moreover, the staining intensity of PDGFRA born positive correlation with NR2F1 (Spearman’s r = 0.831, *p* < 0.001) (Figure 7m). Survival analysis illustrated that highly expressed NR2F1 revealed a significant link with adverse OS (*p* < 0.001), PFS (*p* = 0.004) (Figure 7n,o). Taken together, these findings suggested that enhanced NR2F1 expression was closely linked with infiltration of immunosuppressive CAFs. Based on this, we hypothesized that NR2F1 might contribute to immunotherapy resistance. To verify this, the potential ICB response was predicted utilizing the TIDE algorithm. As shown in Figure 8a,b, the high NR2F1 expression group in TCGA and ICGC OC datasets had significantly elevated TIDE scores (both *p* < 0.001). The expression of NR2F1 was also increased in anti-PD-L1 nonresponders in the IMvigor210 cohort (*p* = 0.0043) and Wolf cohort (*p* = 0.016) (Figure 8c,d). The area under the receiver-operating characteristic curve (AUC) values were 0.617 and 0.671, respectively, revealing a good ability of NR2F1 to discriminate between anti-PD-L1 responders and non-responders (Figure 8e,f).

## 4. Discussion

Ovarian cancer, a life-threatening gynecological malignancy, yields a 5-year survival rate of nearly 47%, a number that has been invariable throughout the past two decades [42]. Clinical studies have evidenced that resistance toward platinum-based therapies is an important factor in impeding the management of OC and causes a poor long-term prognosis [43]. Thereby, the determination of novel targets to overcome platinum resistance is a practical modality for OC treatment. In this study, we detected a novel gene, NR2F1 in platinum-resistant OC cells and tissues. In comparison with the sensitive cells and tissues, NR2F1 expression was increased in platinum-resistant cells and tissues. Furthermore, OC patients with higher NR2F1 expression had inferior prognoses. Univariate and multivariate analysis implied that NR2F1 could predict adverse prognosis independently. Notably, NR2F1 was found to be positively correlated with pathological stages of OC, and could also influence a cellular response to platinum treatment. A highlight of our work is predicting the relevant mechanisms through which NR2F1 regulates the platinum resistance in OC. Based on GSEA analysis, we found that the DEGs between platinum-sensitive and resistant tissues were significantly enriched for activating EMT. The TGFβ pathway, involved in EMT, was also enriched via GO analysis. TGFβ-1, a critical regulator in TGFβ signaling cascades, was overexpressed in platinum-resistant tissues and selected as a possible downstream target of NR2F1. Afterwards, through experimental verification, we substantiated that NR2F1 led to platinum resistance by inducing EMT by elevating TGFβ-1 expression. More importantly, the analysis of the immune infiltration showed that NR2F1 showed the highest association with CAF infiltration, thus, shaping an immunosuppressive microenvironment. High NR2F1 expression was further found to be correlated with poor response to immune checkpoint blockades, such as anti-PD-L1. ROC curves showed the good capacity of NR2F1 in distinguishing anti-PD-L1 responders and nonresponders. From a new perspective of NR2F1 and ovarian microenvironment regulation, our data can provide evidence for developing more effective chemotherapy, and immunotherapy plans for patients with OC.

Up to now, BRCA1/2, TP53, and ARID1A mutations are commonly used in the diagnosis and prognosis predictions of OC patients. BRCA mutations cause homologous recombination defects (HRD) and are associated with improved sensitivity toward DNA-damaging therapies including platinum chemotherapy in OC patients [44]. TP53 is well-recognized as a guardian of the human genome [45], and its mutation leads to abnormal cell growth and other oncogenic functions in human cancers, including OC [46]. The ARID1A mutation was reported to promote OC carcinogenesis through the PI3K/AKT pathway [47]. Our study showed no correlation between NR2F1 expression and BRCA1/2, TP53, and ARID1A mutations, indicating that NR2F1 may be considered an independent character of platin response prediction in OC patients. 

NR2F1 is an established biomarker of tumor dormancy [48]. Studies have reported inconsistent results of NR2F1 affecting cell proliferation in different cancer types. Expression of NR2F1 was lower in breast tumors with high Ki67 expression and proliferation scores [49]. NR2F1-AS1 was overexpressed in the dormant mesenchymal-like breast cancer stem-like cells and favored tumor dissemination but diminished proliferation in lungs through up-regulating NR2F1 expression [12]. Silencing NR2F1 attenuates the proliferation capacity of pancreatic cancer cells [50]. However, the effect of NR2F1 on OC proliferation remains unknown. Our study firstly found that NR2F1 was not correlated with proliferative marker Ki67 in OC, and NR2F1 knockdown did not affect cell proliferation in response to cisplatin. EMT is believed to elevate cancer cells’ capacities to invade and migrate [51]. Accumulative studies have reported that NR2F1 serves a critical function in EMT. For instance, Pakdel and co-workers proved that reinforced expression of NR2F1 in breast cancer cells might contribute to losing the epithelial phenotype and acquiring the mesenchymal characteristics [11]. Similarly, Jiang et al. identified that the dietary supplement ProstaCaid™ (PC) could repress cell migration and invasiveness in prostate cancer by reducing NR2F1 levels, thus, inhibiting metastatic behavior [52]. Furthermore, hypoxia-induced NR2F1-AS1 expression directly augmented NR2F1 levels to favor pancreatic cancer cell migration and invasion through up-regulating AKT/mTOR signaling [50]. Simultaneously, NR2F1 was capable of modulating the drug resistance of cancer cells. Based on the report from Huang’s group, NR2F1-AS1, the expression of which in oxaliplatin-resistant tissues was augmented, could target ABCC1 by sponging miR-363 and confer resistance to hepatocellular carcinoma [53]. CRISPR deletion of NR2F1 was able to restore enzalutamide sensitivity in androgen receptor (AR)-positive prostate cancer LNCaP cell with chromodomain helicase DNA-binding protein 1 (CHD1) knockdown [54]. Consistent with prior literature, we discovered elevated NR2F1 expression in platinum-resistant OC tissues, and silencing of NR2F1 enhanced sensitivity to cisplatin in OC cell lines by promoting migration and invasion. 

TGF-β1, a multipotent cytokine in the TGF-β family, is overexpressed in OC patients’ serum [55]. Increased expression of TGF-β1 has been described as a potent inducer of EMT [56] and an augmented metastatic potential in OC cells [57]. Other studies also showed that TGF-β1 is related to sensitivity to chemotherapeutic agents. For example, TGF-β1 mRNA expression was significantly diminished in tissues of OC patients with increased sensitivity to paclitaxel and carboplatin than in those with reduced sensitivity, and patients with lower TGF-β1 expression tended to have better outcomes [58]. Consistently, TGF-β1 levels were higher in OC cisplatin-resistant cells, and targeting TGF-β1 using SB431542, a TGF-β1 inhibitor, could overcome cisplatin resistance [59]. In our study, remarkably upregulated TGF-β1 was observed in platinum-resistant OC tissues and positively linked with the levels of NR2F1. In particular, we discovered that the external addition of TGF-β1 has attenuated the inhibitory effect of NR2F1 knockdown on the EMT-related factors, cell invasion, and migration. These results indicated that NR2F1 was involved in EMT through targeting TGF-β1 which conferred cisplatin sensitivity in OC cells.

Accumulative studies have revealed that cancer-associated fibroblasts are essential components of the tumor microenvironment (TME), promoting primarily the maintenance, progression, metastasis and therapeutic resistance of OC [60]. Moreover, CAFs are also the main contributors to creating an immunosuppressive microenvironment [61]. Targeting CAF in the ovarian microenvironment may yield alternative therapeutic strategies. PDGFR signaling serves critical role in activating CAFs, and the blockage of PDGFR could impede the recruitment of CAFs to the tumor site, which ultimately repressed cancer growth and metastasis [62,63]. In this paper, NR2F1 was evidenced to be positively associated with CAF infiltration and CAF markers such as PDGFRA and PDGFRB. In particular, IHC analysis illustrated that both NR2F1 and PDGFRA were overexpressed in OC metastatic tissues, and a strongly positive relationship between these two genes was confirmed. Moreover, stromal scores in the TCGA cohort showed that high stromal infiltration could be suggestive of increased NR2F1 expression. These findings indicated that the functional role of NR2F1 might correlate with the modulation of CAFs. TGFβ-1 is acknowledged to be pivotal signaling mediating the transformation of normal fibroblasts into CAFs [64,65]. Activation of TGFβ signaling is responsible for the tumor-promoting action of CAFs, and TGFβ inhibition contributes to the reduction in CAF numbers or CAF activation, exerting anti-cancer effects [66]. Intriguingly, previous studies also demonstrated that the poor response to PD-L1 blockade was attributed to active TGFβ signaling in CAFs [36]. In line with this, we found that NR2F1 could elevate TGFβ-1 signaling. Moreover, high NR2F1 expression was correlated with a poor response to immune check blockades such as anti-PD-L1. According to findings and evidence from scientific literature, we proposed that NR2F1 could also promote the activation of CAF by the induction of TGFβ signaling, thus, conferring resistance to OC. However, in-depth exploration built on grafted tumors and patients’ specimens is greatly required.

Although this study included analyses on human patient samples and in vitro experiments, additional mice models injected with NR2F1 knockdown cell lines or using NR2F1 knockout mice are required to further investigate the intrinsic mechanisms of NR2F1 in OC.

## 5. Conclusions

In summary, this study demonstrated that NR2F1 is an upregulated gene in OC platinum-resistant tissues. The augmented expression of NR2F1 indicated dismal prognoses in OC patients. The mechanism by which NR2F1 confers platinum sensitivity in OC may be associated with the induction of EMT through the elevation of TGF-β1. Of importance, our study also discussed that NR2F1 could promote the activation of CAFs by inducing TGFβ signaling, which is associated with a poor immunotherapeutic response. Our study indicated that NR2F1 represents a promising therapeutic target for OC patients to overcome resistance toward chemotherapy and immunotherapy.

## Figures and Tables

**Figure 1 cancers-14-04639-f001:**
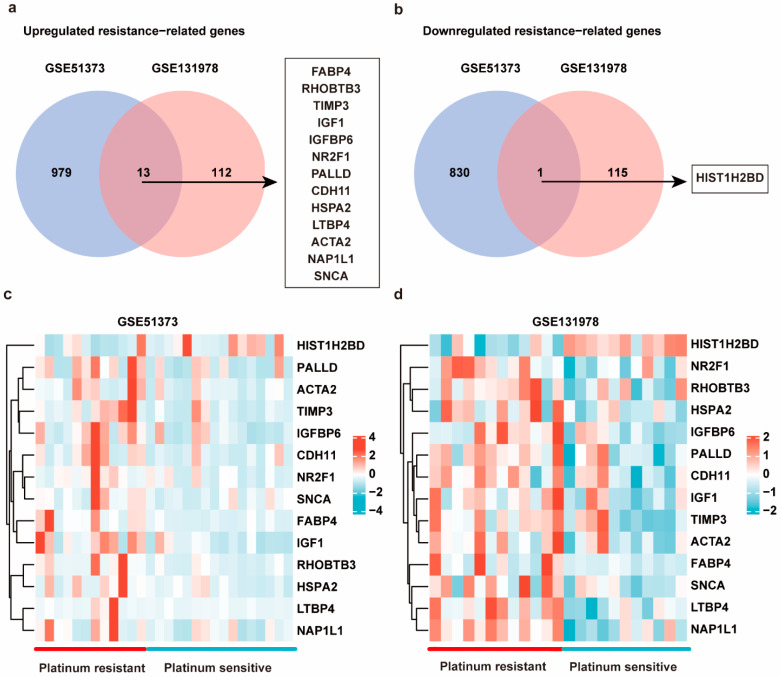
The common differential expressed genes (co-DEGs) associated with platinum resistance between GSE51373 and GSE131978. (**a**) Thirteen co-upregulated platinum-resistant-related genes (FABP4, RHOBTB3, TIMP3, IGF1, IGFBP6, NR2F1, PALLD, CDH11, HSPA2, LTBP4, ACTA2, NAP1L1, and SNCA) were identified in ovarian cancer (OC) tissues. (**b**) One co-downregulated gene (HIST1H2BD) was identified in OC tissues. (**c**,**d**) Heatmaps of 14 co-DEGs between platinum-resistant and sensitive tissues.

**Figure 2 cancers-14-04639-f002:**
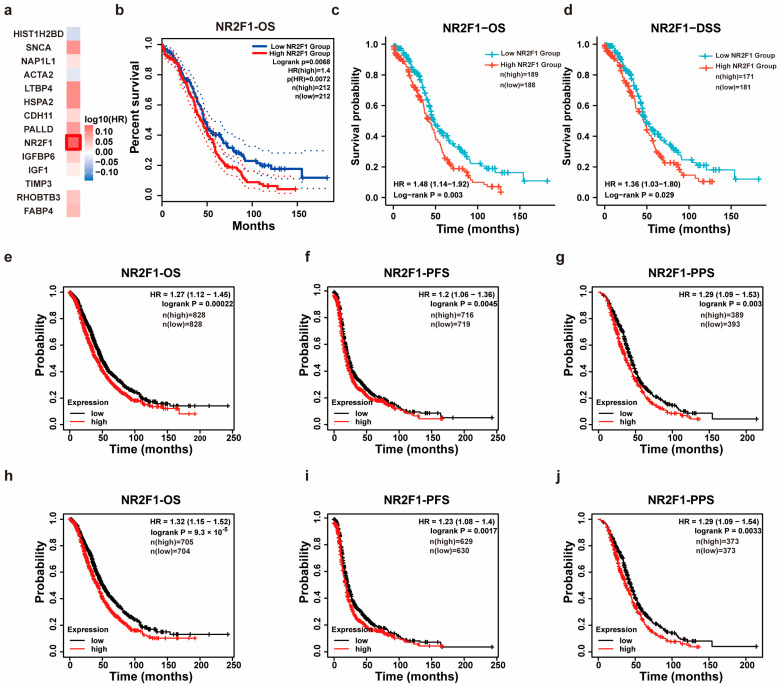
Prognostic significance of NR2F1 in OC. (**a**,**b**) Relationship between co-DEGs and overall survival (OS) by GEPIA2. Kaplan-Meier curve revealed the association of NR2F1 expression with OS. (**c**,**d**) The link of NR2F1 expression with OS, and disease-free survival (DSS) is based on Xiantao tool. (**e**–**j**) Kaplan-Meier survival analyses of NR2F1 on OS, progression-free survival (PFS) and post-progression survival (PPS) in OC patients (**e**–**g**), and in OC patients receiving platinum treatment (**h**–**j**).

**Figure 3 cancers-14-04639-f003:**
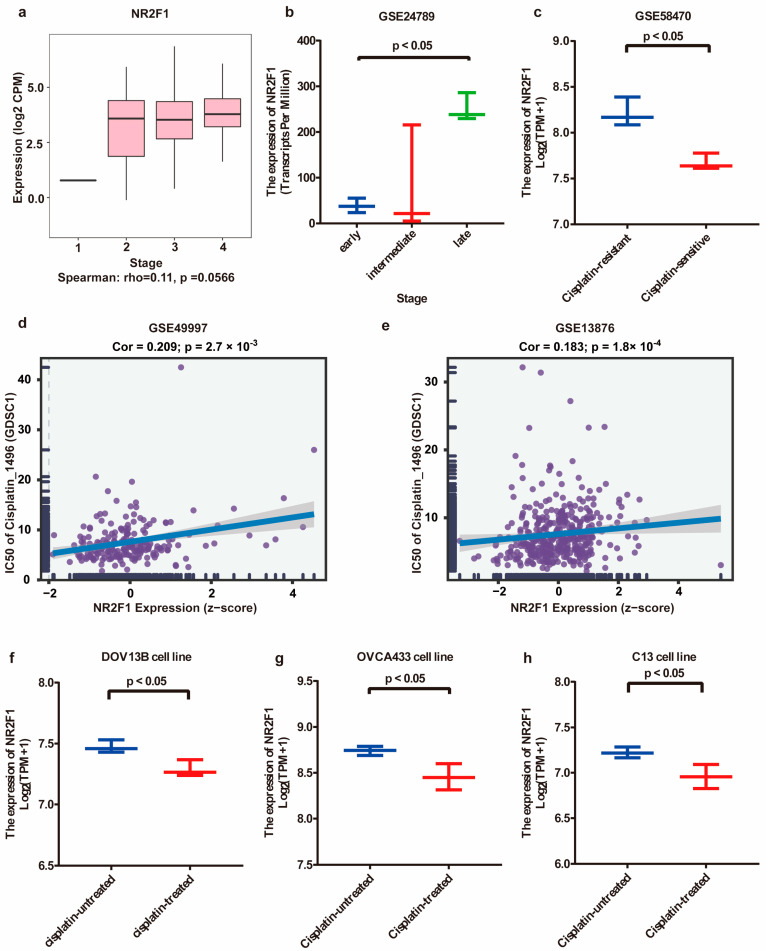
The association of NR2F1 with OC progression and cellular response to platinum treatment. (**a**) NR2F1 expression tendency in OC tissues at different clinical stages by TISIDB. (**b**) NR2F1 expression tendency in mouse ovarian surface epithelial (MOSE) cells at different stages based on GSE24789. (**c**) Confirmation of the elevated NR2F1 expr-ession in cisplatin-resistant OC cell lines using GSE58470. (**d**,**e**) Scatter plots of the association between NR2F1 expression and IC50 values of cisplatin in GSE49997 and GSE13876 datasets. (**f**–**h**) Changes in NR2F1 expression levels in OC DOV13B, OVCA433 and C13 cell lines following cisplatin treatment.

**Figure 4 cancers-14-04639-f004:**
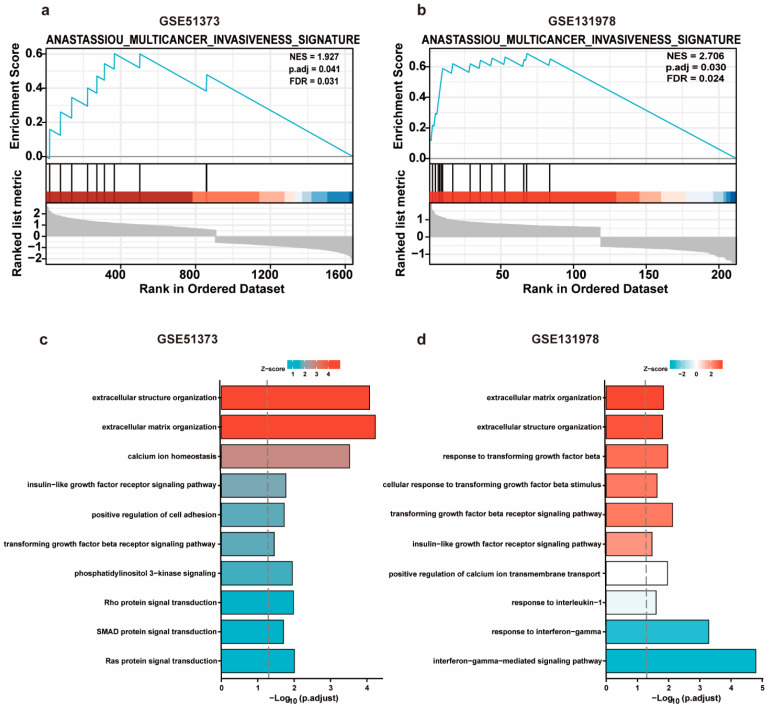
GSEA analysis and GO enrichment analyses of platinum-resistant-associated DEGs. (**a**,**b**) GSEA analysis of DEGs in GSE51373 and GSE131978. (**c**,**d**) GO (Biological Process) enrichment analyses of DEGs in GSE51373 and GSE131978. Only pathways of interest showing a *p*-value < 0.05 are presented.

**Figure 5 cancers-14-04639-f005:**
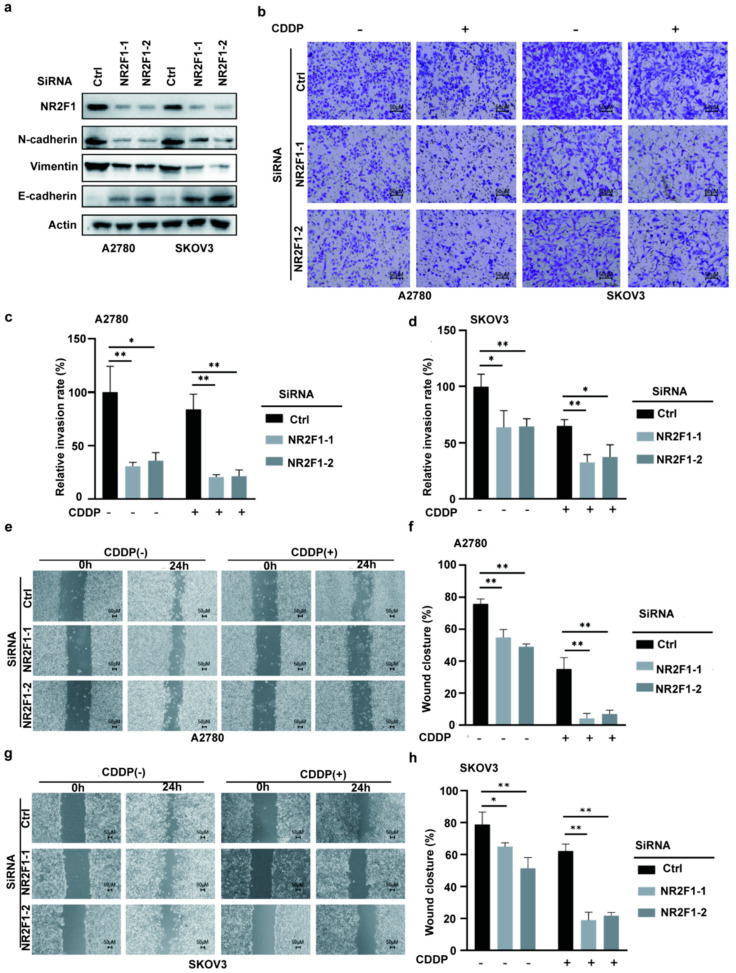
Silencing NR2F1 inhibits EMT. (**a**) Western blot analysis of NR2F1, E-cadherin, N-cadherin and vimentin in siNR2F1-transfected A2780 and SKOV-3 cells. The uncropped blots are shown in Appendix A. (**b**–**d**) Examination of the invasiveness of siNR2F1-transfected A2780 and SKOV3 cells without or with cisplatin treatment (20 μM) using transwell assay. (**e**–**h**) Examination of the migration of siNR2F1-transfected A2780 and SKOV3 cells without or with cisplatin treatment (20 μM) using wound-healing assay. CDDP, cisplatin; * *p* < 0.05; ** *p*  <  0.01.

**Figure 6 cancers-14-04639-f006:**
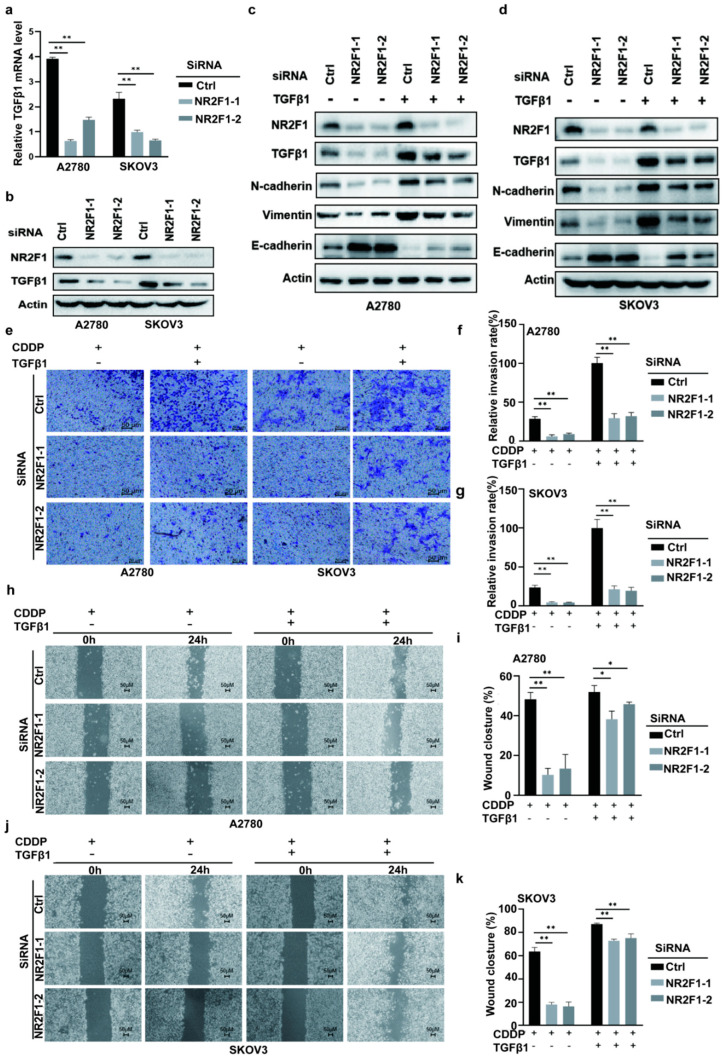
NR2F1 induced epithelial to mesenchymal transition (EMT) via upregulating TGF-β1 expression. (**a**,**b**) Assessment of TGF-β1 expression in siNR2F1-transfected A2780 and SKOV-3 cells using RT-PCR and western blot. (**c**,**d**) SiNR2F1-transfected A2780 and SKOV-3 cells were treated with TGF-β1 (5 ng/mL) or without TGF-β1, and protein expression of NR2F1, TGF-β1, E-cadherin, N-cadherin, and vimentin was determined through western blot. The uncropped blots are shown in Appendix A. (**e**–**g**) After cisplatin (20 μM) treatment, siNR2F1-transfected A2780 and SKOV-3 cells were treated with TGF-β1 (5 ng/mL) or without TGF-β1, and the invasiveness was examined using transwell assay. (**h**–**k**) After cisplatin (20 μM) treatment, siNR2F1-transfected A2780 and SKOV-3 cells were treated with TGF-β1 (5 ng/mL) or without TGF-β1, and the migration was examined using wound-healing assay. CDDP, cisplatin; * *p* < 0.05; ** *p* < 0.01.

**Figure 7 cancers-14-04639-f007:**
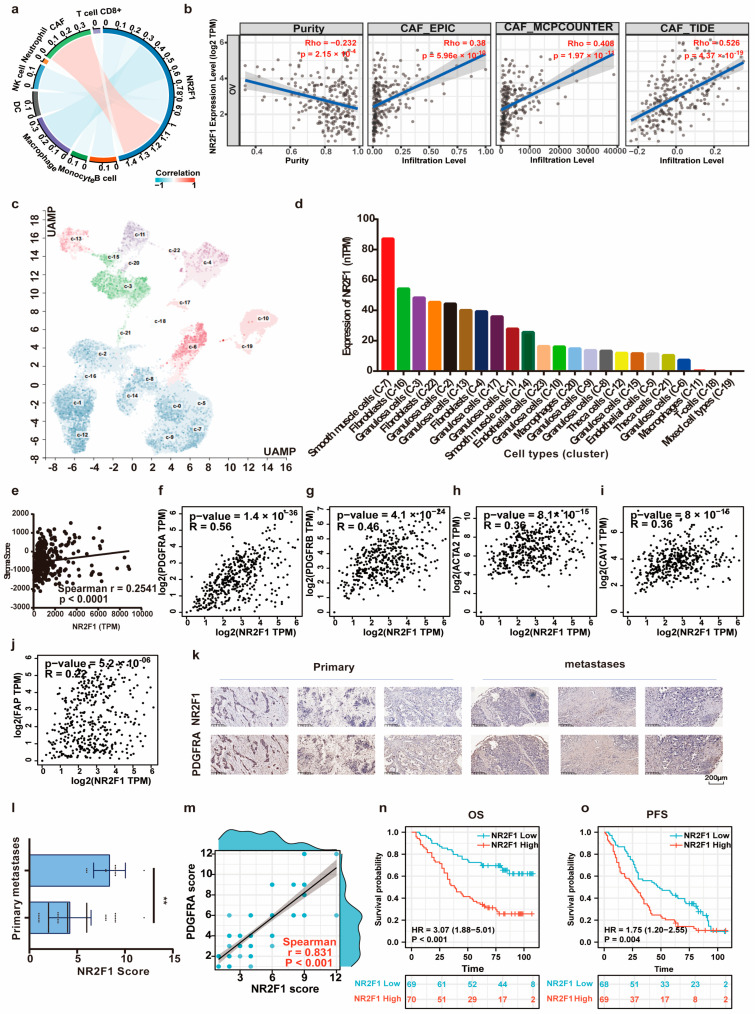
NR2F1 positively linked with cancer-associated fibroblast (CAF) infiltration. (**a**) The connection between NR2F1 and multiple kinds of tumor-infiltrating immune cells in OC by XCELL algorithm. (**b**) The association between NR2F1 and the infiltration of CAFs in OC according to EPIC, MCPCOUNTER, and TIDE algorithms. (**c**,**d**) The UMAP plot and the bar chart visualized NR2F1 expression in the single-cell types clusters in ovary tissues. (**e**) The relationship between NR2F1 with stromal score generated utilizing ESTIMATE algorithm, according to TCGA OC dataset. (**f**–**j**) The relationship between NR2F1 and markers of CAFs including PDGFRA, PDGFRB, ACTA2, CAV1, and FAP in GEPIA2. (**k**) Representative images of NR2F1 and PDGFRA protein expression in metastases and primary OC tissues. (**l**) NR2F1 was upregulated in metastases versus primary OC tissues. ** *p* < 0.01. (**m**) The relationship between NR2F1 and PDGFRA in all OC tissues. (**n**,**o**) Kaplan-Meier survival analyses of NR2F1 on OS, PFS in all OC patients.

**Figure 8 cancers-14-04639-f008:**
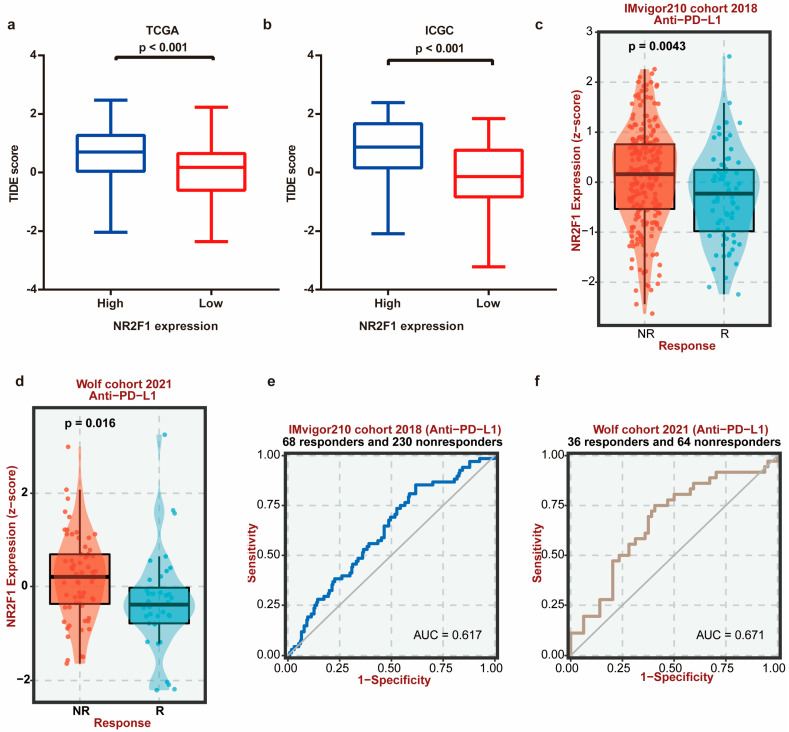
High NR2F1 expression predicts poor immunotherapeutic response. (**a**,**b**) High NR2F1 expression group displayed higher TIDE scores than the low-expression group in TCGA and ICGC OC datasets. (**c**,**d**) NR2F1 expression was increased in anti-PD-L1 responding patients relative to non-responding patients according to IMvigor210 cohort and Wolf cohort. NR: non-responders; R: responders (**e**,**f**) receivers operating characteristic (ROC) curves of NR2F1 for patients in IMvigor210 cohort and Wolf cohort.

**Table 1 cancers-14-04639-t001:** Univariate and Multivariate analysis of risk factors and OS in TCGA.

Variables	HR (95% CI)	*p*-Value
**Univariate analysis**		
NR2F1 (low vs. high)	1.486 (1.145–1.929)	0.003
FIGO stage (Stage I&Stage II vs. Stage III&Stage IV)	2.115 (0.938–4.766)	0.071
Primary therapy outcome (PD&SD vs. PR&CR)	0.301 (0.204–0.444)	<0.001
Race (Asian vs. Black or African American)	1.302 (0.437–3.882)	0.636
Race (Asian vs. White)	0.785 (0.290–2.127)	0.634
Age (≤60 years vs. >60 years)	1.355 (1.046–1.754)	0.021
Histologic grade (G1&G2 vs. G3&G4)	1.229 (0.830–1.818)	0.303
Anatomic neoplasm subdivision (Bilateral vs. Unilateral)	0.953 (0.705–1.289)	0.757
Venous invasion (No vs. Yes)	0.896 (0.487–1.649)	0.723
Lymphatic invasion (No vs. Yes)	1.413 (0.833–2.396)	0.200
Tumor residual (NRD vs. RD)	2.313 (1.486–3.599)	<0.001
**Multivariate analysis**		
NR2F1 (low vs. high)	1.439 (1.054–1.963)	0.022
Primary therapy outcome (PD&SD vs. PR&CR)	0.287 (0.189–0.435)	<0.001
Tumor residual (NRD vs. RD)	2.301 (1.390–3.810)	0.001

HR, hazard ratio; CI, confidence interval; OS, overall survival; FIGO, Federation International of Gynecology and Obstetrics; PD, progressive disease; SD, stable disease; PR, partial response; CR, complete response; RD, recurrent disease; NRD, on-recurrent disease.

**Table 2 cancers-14-04639-t002:** Association between expression of NR2F1 and clinicopathologic characteristics in OC tissues.

Characteristic	NR2F1 High	NR2F1 Low	*p*-Value
n	69	68	
Age, mean ± SD	52.37 ± 11.66	48.72 ± 10.15	0.054
Pathologic stage, n (%)			<0.001
Stage I	3 (2.2%)	5 (3.6%)	
Stage II	8 (5.8%)	23 (16.8%)	
Stage III	34 (24.8%)	33 (24.1%)	
Stage IV	24 (17.5%)	7 (5.1%)	
T, n (%)			0.003
T1	3 (2.2%)	5 (3.6%)	
T2	8 (5.8%)	23 (16.8%)	
T3	58 (42.3%)	40 (29.2%)	
N, n (%)			<0.001
N0	40 (29.2%)	60 (43.8%)	
N1	29 (21.2%)	8 (5.8%)	
M, n (%)			0.001
M0	45 (32.8%)	61 (44.5%)	
M1	24 (17.5%)	7 (5.1%)	
ki67 intensity, median (IQR)	1.75 (1.5, 2)	1.75 (1, 2)	0.140
ki67 extent, median (IQR)	0.3 (0.11, 0.44)	0.2 (0.05, 0.4)	0.057
EGFR intensity, median (IQR)	0.5 (0.5, 1)	0.5 (0.5, 1)	0.614
EGFR extent, median (IQR)	0.55 (0.12, 0.92)	0.5 (0.1, 0.72)	0.450

## Data Availability

The data presented in this study are available in publicly accessible repositories. GSE51373, GSE131978, GSE24789 and GSE58470 datasets are available in GEO (http://www.pubmed.com/geo, accessed on 26 March 2022); HPA, GTEx, and FANTOM5 datasets are available in HPA database (https://www.proteinatlas.org/, accessed on 1 July 2022). TCGA dataset and ICGC dataset of OC are available in Home of Clincal BioInformatics (https://www.aclbi.com/static/index.html#/, accessed on 25 June 2022).

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
