# Peer review of "NR2F1 Regulates TGF-β1-Mediated Epithelial-Mesenchymal Transition Affecting Platinum Sensitivity and Immune Response in Ovarian Cancer"

_cancers, 2022, doi:10.3390/cancers14194639_

Round 1
Reviewer 1 Report
The manuscript “NR2F1 regulates TGF-β1-mediated epithelial-mesenchymal transition affecting platinum sensitivity and immune response in ovarian cancer” is a research article that showed how NR2F1 knockdown suppresses cell invasion and migration with or without cisplatin through EMT pathway in ovarian cancer with a potential impact on TGF-β1 signaling. The manuscript is well written, although few typos are present. To be considered for publication, authors should address the following issues:
1. In the introduction authors should prove a short background on mechanisms of action of cisplatin. It is important to specify that within cancer cells, cisplatin binds to the N7 of purine bases leading to DNA structural damage, with a subsequent cell division block and further induction of the apoptotic program. Besides DNA damage, cisplatin can also lead to the induction of cell death by promoting ROS production. Indeed, oxidative stress represents a fundamental condition through which cisplatin cytotoxicity is exerted, and massive ROS release results in the apoptotic pathway activation (PMID: 33297311).
2. The introduction regarding TGFB1 desearves to be improved in order to highlight its pleotrophic actions. In fact, it desearves to be added that, in addition to EMT, TGFB1 can also induce collagen production (PMID: 32006713) favoring cancer cell motility and progression.
3. Line 62: Authors should highlight that hormones (especially estrogens and progesterone) play a key role in ovarian cancer progression (as recently reviewed PMID: 35453348) further highlighting the importance of their findings.
4. Line 164: please correct CO2 with number “2” in lowercase.
5. How was cisplatin dissolved prior treatment?
6. Line 200: 1.2×106 cells/mL is wrong. “6” must be uppercase.
7. In wound healing assay, what was the concentration of FBS in the medium used for the assay?
8. Line 212: “-ΔΔct” must be with uppercase.
Reviewer 2 Report
Ovarian cancer (OC) is the 5th leading cause of cancer related deaths among women. One of the main reasons for low survival rate among OC patients is development of chemoresistance. So understanding mechanisms underlying the development of chemoresistance is very important. Here, the authors have shown that platinum resistant cells and tissues have high expression of NR2F1 and patients with high NR2F1 expression have poor prognosis. They found that NR2F1 positively corelated with pathological stages of OC. They demonstrate that NR2F1 regulates EMT in OC via induction of TGF-b1 and knocking down NR2F1 inhibits EMT while treatment of NR2F1 KD cells with GF-b1 rescues the EMT inhibition. Furthermore, they also show that NRF2F1 promotes an immunosuppressive phenotype in OC by positively regulating the infiltration of CAFs. The manuscript is interesting but can be improved based on following comments –
Major and Minor Comments -
1. In the abstract, “Moreover, gene set enrichment analysis …. were applied ..” Here “used” is a better word than “applied”.
2. In the abstract, “We used bioinformatic analysis …. differential expressed” there is a typo. It should be “differentially”.
3. In the introduction “In particular, evidence from accumulating ….containing hormones” should be “including hormones”.
4. In the introduction “ OC, with heavily invasiveness..” should be “OC cancer which is highly metastatic ..rephrase the sentence”
5. Do patients with high NR2F1 expression fall in a specific mutation category? E.g do they all have a BRCA1 mutation or ARID1A mutation? This will be informative to see if NR2F1 is useful for patients with specific genetic mutation?
6. In section 3.3, the authors say “Through analysis of HPA, GTEx, and FANTOM5 …… ovary tissues. Do the authors mean normal ovary tissue?
7. In section 3.4, the line “ In conclusion… NR2F1 in platinum resistance “ is confusing. Please rephrase this sentence.
8. NR2F1 KD affects invasion and migration of cells in response to cisplatin. Does NR2F1 KD also affect proliferation of cells in response to cisplatin?
Reviewer 3 Report
Basic Reporting and Comments
In this manuscript authors have identified a role of nuclear receptor NR2F1 in regulating TGF-beta mediated EMT which affects the drug response and immune response in ovarian cancer. They have used publicly available genomic datasets to find the correlation between increased expression of NR2F1 and the response to platinum therapy. They have found the positive correlation between NR2F1 expression and infiltration of immune suppressive cancer associated fibroblasts. Also, they have seen knockdown of NR2F1 in ovarian cancer cell lines suppresses the invasion and migration thereby making the cells more sensitive to cisplatin therapy.
1. Dose dependence experiment done with cisplatin 0-20 micromolar should be included in supplementary data. As these concentrations seem high to be used as a drug treatment. Also, in the experiments done in figure 6, they have used cisplatin as 5 ng/ml which does not relate to these concentrations. Please explain the discrepancy. Also, what is the concentration of cisplatin used in Figure 5??
2. It is ideal to give the full catalog numbers of the antibodies in the methods sections for the audience who are going do similar studies in future.
3. Three are typo errors while writing the cell numbers plated in line 192 and 200.”6” should in superscript.
4. siRNAs are not considered ideal for doing invasion and migration experiments. shRNAs are a good choice as the effect of invasion and migration needs to be checked for atleast 72 hours. Also, if you look at Fig 5A, siRNA doesn’t seem very efficient in inhibiting the NR2F1 expression.
5. 0 and 24 hour pictures for scratch assay is too less. Ideally the scratch is made with woundmaker in the commercially available incucyte instrument to make a homogenous/ similar scratch all over in every well. The pictures should be taken every 4 hours and the experiment should be done for atleast 3-4 days. To make a good conclusion.
6. Figure 5D how is the relative invasion rate percentage calculated? Please explain in detail. Number of cells invaded were counted or area? It would be helpful
7. Figure 6B Instead of western blot, ELISA would be a good choice for secretory cytokine like TGF-beta. Also, cell line labelling is missing in this figure panel.
8. How much TGF-beta conecntration is used in Figure 6 c-K ?? Also, although cisplatin treatment info is written in legends for figure 6 c-d cisplatin labelling needs to be added in the figure panels for better understanding.
9. For better supporting the fact that NR2F1 higher expression leads to infiltration of CAFs. A simple coculture experiment can be done with the NR2F1 KD cells and undifferentiated monocytes. As NR2F1 KD cells will secrete less TGFB1 which can lead to less of immunosuppressive M2 macrophage differentiation and more of inflammatory M1 macrophages. TGF beta is known to skew M2 macrophage differentiation. This experiment can be a mimic of tumor microenvironment.
10. Although the study includes informatic analyses on human patient samples which is a plus to study, still it’s a correlative study, mice model injected with NR2F1 KD cell lines or using NR2F1
KO mice can better answer the mechanistic basis explained in this study and justify that NR2F1 can be used as a marker for treating patients with ovarian cancer.
